# Use of Outpatient Health Services by Mexicans Aged 15 Years and Older, According to Ethnicity

**DOI:** 10.3390/ijerph20043048

**Published:** 2023-02-09

**Authors:** Blanca Estela Pelcastre-Villafuerte, Leticia Avila-Burgos, Sergio Meneses-Navarro, Nadia Cerecer-Ortiz, Julio César Montañez-Hernández

**Affiliations:** 1Centre for Health Systems Research, National Institute of Public Health, Cuernavaca 62100, Mexico; 2Centre for Health Systems Research, National Institute of Public Health-CONACyT, Cuernavaca 62100, Mexico

**Keywords:** outpatient services, use, ethnicity

## Abstract

The aim of this study was to estimate the prevalence of health needs and use of outpatient services for indigenous (IP) and non-indigenous (NIP) populations aged ≥15 years, and to explore the associated factors and types of need. A cross-sectional study was conducted based on the 2018-19 National Health and Nutrition Survey. The population aged ≥15 years who had health needs and used outpatient services was identified. Logistic models were developed to explore the factors underlying the use of outpatient services. For both populations, being a woman increased the likelihood of using health services, and having health insurance was the most important variable in explaining the use of public health services. Compared to the NIP, a lower proportion of IP reported health needs during the month prior to the survey (12.8% vs. 14.7%); a higher proportion refrained from using outpatient services (19.6% vs. 12.6%); and a slightly higher proportion used public health services (56% vs. 55.4%). For the NIP, older age and belonging to a household that had received cash transfers from a social program, had few members, a high socioeconomic level, and a head with no educational lag, all increased the likelihood of using public health services. It is crucial to implement strategies that both increase the use of public health services by the IP and incorporate health-insurance coverage as a universal right.

## 1. Introduction

Universal health coverage cannot be achieved unless all social groups -especially those who have systematically suffered from structural inequality- enjoy effective and equitable access to health services [1,2]. Effective access begins when health needs are recognized and prompt either individuals or populations to seek care. It includes the availability of resources required to provide health services; geographical, economic, organizational and cultural accessibility; acceptability; adequate technical quality, dignified treatment, equity and financial protection [3,4]. Assessing the use of services constitutes the most fundamental way to measure access to healthcare.

However, in all societies, specific sectors of the population encounter structural barriers to the exercise of their social rights, including health protection. In Mexico, the indigenous population (IP) has historically exhibited the most adverse well-being indicators, such as barriers to accessing health services [5]. According to the 2020 Population and Housing Census, indigenous language speakers aged ≥3 years represent approximately 6.1% of the total Mexican population [6]. Indicators for their living conditions show systematic disadvantages relative to the non-indigenous population (NIP). For example, over 70% of indigenous people are immersed in poverty and suffer from high levels of marginalization [7]; they face greater obstacles in accessing basic education [8]; exhibit the highest maternal and infant mortality rates; and have a life expectancy at birth that is up to 20 years lower than the national average [9]. This notwithstanding, their use of public health services is lower than that of their non-indigenous counterparts [10].

Studies have shown that ethnicity does not in itself explain the lower use of health services among the IP, but is related to other conditions such as poverty, fewer years of schooling [11,12], insufficient availability of health services in certain regions [13], the lack of cultural sensitivity in the delivery of healthcare services [14,15] and even discriminatory attitudes and practices, as well as distinct forms of violence and mistreatment in health institutions [16,17,18].

Only a small percentage of the Mexican IP is affiliated with a health insurance scheme. This is largely because the national health system is segmented by employment status [19]. In 2020, 36.5% (45.99 million) of the poorest stratum of the population (self-employed and unemployed individuals, and workers in the agricultural and informal sectors, including the IP) were excluded from Social Security (SS) and received care through schemes and programs funded by federal government agencies [20,21]. Far from being ameliorated, segmentation has deepened in recent years. According to the National Council for the Evaluation of Social Development Policy (CONEVAL by its Spanish initials), the percentage of the Mexican population suffering from deprivation caused by lack of access to health services increased from 16.2% in 2018 to 28.2% in 2022. During the same period, lack of access increased by as much as 31.7 percentage points (from 25.6% to 57.3%) for those living in extreme poverty and by 16.8 percentage points (rising from 13.7% to 30.5%) for those living in rural areas [22]. Most of the IP is trapped in these circumstances [23]. Having no affiliation with any health insurance scheme is one of the first barriers encountered by the IP during their attempts to exercise their right to health protection. According to the 2020 Population and Housing Census, 23.1% of the IP lacks health insurance coverage, while 76.9% benefits from health insurance schemes. Of these, 99.2% enjoys access to health services financed by the federal government through either the Institute of Health for Well-being or the IMSS-Bienestar program, and the remaining 0.8% is affiliated with a SS scheme [20].

The 2021 National Health and Nutrition Survey [24,25] revealed that among the 11.4% of the Mexican population reporting a health need during the month prior to the survey, 87% received care, but less than half (41%) received care from a public provider [26]. It should be noted that, in recent years, care obtained from public services has decreased. This is reflected in higher out-of-pocket spending on health among Mexican households and a heavier financial burden for the poorest households [12].

These data contextualize the unfavorable conditions that the IP has faced in accessing and using health services. Given the fundamental role of public services in achieving universal health coverage and financial protection, we undertook the present study to analyze the differences between the IP and NIP in the use of public health services when faced with a health need, a phenomenon known as the “cascade of care” [12,26]. We hypothesized that the results would serve as an indicator of progress towards greater access and use of public health services by the IP. The purpose of our work was thus to estimate the prevalence rates of reported health needs and use of public outpatient health services among IP and NIP members aged ≥15 years. We identified the factors underlying and the types of health need motivating use of services among these populations.

## 2. Materials and Methods

### 2.1. Study Design, Data Collection and Population

We conducted a cross-sectional study based on the 2018-19 National Health and Nutrition Survey (ENSANUT). The survey used a probabilistic multistage design, stratifying and analyzing data by conglomerates at the national, urban, and rural levels. Methodological details have been previously published [25]. We used the survey questionnaires concerning households and outpatient health-service users to obtain data on sociodemographic characteristics, health status, use of outpatient health services, and health needs motivating the quest for care.

Although the 2018–2019 ENSANUT collected data from 158,044 individuals (N = 126.47 million), we analyzed only those aged ≥15 years, given the greater participation of the IP in the agricultural and informal sectors, both of which are associated with an earlier initiation into the field of labor [27,28,29]. We classified household members relative to their indigenous or non-indigenous ethnicity. For this purpose, we used the definition of the National Institute of Indigenous Peoples (INPI by its Spanish initials), according to which a household is considered indigenous when the head, his/her spouse, a parent, grandparent, great grandparent or the equivalent on the spouse’s side identifies as an indigenous language speaker [30]. Under this criterion, among the total sample population aged ≥15 years, which amounted to 116,730 individuals (N = 94.15 million), 9371 (N = 6.72 million), or 7.1%, were considered indigenous (IP), and the remaining 107,359 (N = 87.42 million) were considered non-indigenous (NIP).

### 2.2. Variables of Interest

#### 2.2.1. Use of Health Services

Consistent with previous studies [12,26], we adopted the care-cascade approach. We first identified individuals who had reported a health need and then estimated the proportion who had used outpatient healthcare services. Use of services flows from the interaction among supply and demand factors [31]. Our analysis considered use from the demand perspective; that is, based on the individual and household characteristics that influence the decision to use healthcare services.

The questionnaire for outpatient service users began by identifying individuals with at least one health need, and then explored their use of outpatient health services for treatment. Two questions were formulated for this purpose: (1) “In the last month, have you had any health problems related to a disease, a physical injury, an accident or an assault?” and (2) “In the last two weeks, have you requested a consultation, not requiring hospitalization, for a disease or disease control, an injury or an accident?” After identifying those who answered affirmatively to either question, our initial sample recorded 1265 (N = 859,265) indigenous and 16,504 non-indigenous individuals aged ≥15 years with a health need. These were then classified, establishing which ones had received care from an institutional healthcare provider and whether the type of provider was public or private. Finally, indigenous and non-indigenous people aged ≥15 years who had received care from an institutional public healthcare provider during the month preceding the survey were defined as public outpatient service users. Conversely, indigenous and non-indigenous people ≥15 years of age with a health need who did not receive care; received care from relatives, friends, neighbors or alternative personnel; or depended on a drugstore for treatment, were defined as non-users. Figure 1 illustrates our sample selection process.

Public healthcare providers included SS institutions and other government agencies devoted to those without SS coverage; private healthcare providers included private doctors’ offices and doctors’ offices adjacent to private pharmacies (DAPPs). Institutional health personnel included physicians, nurses, nutritionists, dentists and health assistants.

Evidence has shown that the IP has varying preferences for the types of personnel that attend to their health needs [5]. We therefore found it pertinent to identify which types of care provider attended to the needs of the IP and NIP. Institutional as well as alternative and traditional personnel were considered. On this basis, we constructed a variable including three different types of care provider: 0 = no care; 1 = alternative personnel (herbalists, midwives, naturopaths and homeopaths); 2 = general practitioners or specialists; and 3 = other institutional health personnel (nurses, nutritionists, dentists and health assistants).

We began by identifying individuals who were ≥15 years old, pertained to the IP or NIP, and reported having experienced a health need during the month prior to the survey. Of these, we identified those who not only had requested a doctor’s appointment for their health needs but had also received care from institutional healthcare personnel. Lastly, we identified those who had received care from a public health provider, defined as users of institutional public health services. Non-users were defined as individuals who had experienced a health need in the last month but did not receive care from any institutional healthcare provider (whether public or private).

Note: Institutional healthcare personnel included general practitioners and specialists, nurses, nutritionists, dentists and health assistants.

#### 2.2.2. Covariates

We included those covariates that had been associated with the use of health services [12,26,31,32] and were addressed in the ENSANUT Household Questionnaire. They were broken down into individual, household and rural/urban area.

Individual characteristics included age (years); sex (0 = male, 1 = female); schooling (0 = none, 1 = one to six years, 2 = seven to nine years, 3 = ten years and beyond); civil status (0 = without a partner and 1 = with a partner); being employed (0 = no, 1 = yes); and having health insurance (0 = none, 1 = a SS scheme, 2 = a scheme for those without SS coverage, including the Seguro Popular and IMSS Prospera Programs, and 3 = a private-insurance scheme or other).

Household characteristics included the number of household members; educational gaps (0 = the head of household was >15 years old and had <9 years of schooling, and 1 = otherwise [33]); having received a cash transfer or scholarship from either the *Prospera* Social Program (now designated as Benito Juárez Welfare Scholarships) or the Universal Pension Program for Older Adults during the year prior to the survey (0 = no, 1 = yes); and socioeconomic status (SES) divided into tertiles (0 = low, 1 = intermediate, and 2 = high). The SES index was constructed according to the principal components in housing conditions and assets.

Area characteristics included size (0 = rural < 2500 inhabitants, and 1 = urban ≥ 2500 inhabitants); and municipal marginalization level estimated as a continuous variable: the lowest values denoted very high levels, and the highest denoted very low levels [34].

### 2.3. Statistical Analysis

First, we described the individuals aged ≥15 years in the IP and NIP according to their sociodemographic characteristics, and then we determined their care cascades by estimating (1) the prevalence rates of reported health needs and use/non-use of health services by those with reported health needs; (2) the types of healthcare provider attending needs; and (3) the prevalence of service use categorized by the type of care provider and users’ health insurance scheme.

We then explored the differences between the sociodemographic characteristics of the IP and NIP, using the chi-square test for categorical and Student’s *t* test for continuous variables. Finally, we constructed a logistic regression model for each population with use of services as the response variable (0 = non-use and 1 = use). The models estimated the association between covariates and the likelihood of using institutional public outpatient health services. We obtained favorable outcomes from the post-estimation goodness-of-fit tests, indicating that both regression models had been accurately specified (Hosmer–Lemeshow goodness-of-fit >0.05, and variance inflation factor-VIF <10). Estimates were expressed as odds ratios (ORs) with a 95% confidence interval (95% CI). All estimates were adjusted for the complex survey design using Stata 14.0 software [35,36].

The ENSANUT administered a questionnaire module to a sample of outpatient service users in order to explore the factors related to care received and the types of health need that motivated their search for care. From this module, we obtained information for 249 indigenous (N = 146,881) and 3570 non-indigenous (N = 2,571,065) individuals who had reported the type of self-perceived health problem they had experienced. The answer options for this question included a list of 45 diseases, which we grouped into 12 categories. (See Appendix A). We assessed the differences between the two groups of individuals using the chi-square test.

### 2.4. Ethical Aspects

The 2018–2019 ENSANUT was approved by the Ethics, Research and Biosafety Committees of the National Institute of Public Health (Authorization Number 1556). Informed consent forms were signed by all adult participants and written consent was obtained from the parents or guardians of minors. Our study was exempt from the approval protocol because it was based on secondary data from the survey; the data analyzed are publicly available in the survey repository hosted by the National Institute of Public Health, at https://ensanut.insp.mx/ accessed on 16 January 2023, in accordance with the Internal Regulations of the Institute’s Ethics and Research Committees.

## 3. Results

### 3.1. Characterístics of the Indigenous and Non-Indigenous Populations Analyzed

As shown in Table 1, the IP generally experienced less favorable socioeconomic conditions compared to the NIP: the proportion of individuals with an elementary or lower educational level was 25.7 percentage points higher; the proportion of employed individuals was smaller (54.8% vs. 57.4%); and their health insurance schemes mostly consisted of government programs specifically designed for individuals without SS coverage (67.8% vs. 37.2%). More IP households fell within the lower socioeconomic tertiles (70.7% vs. 26.1%), had numerous members (4.8 vs. 4.2, on average), were headed by individuals with educational gaps (73.8% vs. 40.8%), were located in rural areas (49.4% vs. 19.4%), pertained to municipalities with substantial levels of marginalization and had received cash transfers from a government program (34.9% vs. 13.9%).

Figure 2A shows that, compared to the NIP, a smaller proportion of individuals from the IP reported health needs in the last month (12.8% vs. 14.7%) (*p* < 0.01), and a larger proportion reported not having used institutional health services to meet their needs (19.6% vs. 12.6%); this was mainly because they received care from alternative sources (1.4% vs. 0.8%) rather than from health personnel (18.2% vs. 11.8%). In other words, a lower proportion of individuals from the IP with health needs received care from institutional health personnel (80.4% vs. 87.4%), indicating that they received less care from physicians compared to those from the NIP (75.1% vs. 84.4%).

As shown in Figure 2B, no differences emerged between individuals without health insurance coverage from the IP and NIP regarding the types of institution from which they obtained care: both mainly used private healthcare providers. The fact that affiliation with a health insurance scheme has been associated with greater use of health services does not mean that individuals obtain care from their corresponding institutions: regarding SS affiliates, the NIP obtained care more frequently from SS institutions compared to the IP (64% vs. 60%), and the latter more frequently obtained care from institutions aimed at those without SS coverage and private doctors’ offices than did the NIP (6.9% and 27.7% vs. 3.1% and 22.2%). With regard to people affiliated with schemes aimed at individuals without SS coverage, the IP more often received care at SS institutions compared to the NIP (57.5% vs. 52.9%); however, the latter used DAPPs more frequently (15.8% vs. 9.9%). Finally, as expected, a mere 0.1% of the IP vs. 1.1% of the NIP had private health insurance; among the latter, 92% obtained care from private facilities.

First and foremost, Table 2 shows that, among the IP and NIP who used institutional outpatient services, a higher proportion of the former used public services (56% vs. 55.4%) (*p* < 0.01) or received no care from institutional healthcare providers, whether public or private (19.6% vs. 12.6%) (*p* < 0.001). For both populations, being a woman, being affiliated with a health insurance scheme (mainly SS), belonging to a household in an urban area and residing in a municipality with very low levels of marginalization were all associated with greater use of public health services. Conversely, being employed reduced such use.

Additionally, for the NIP, we identified other characteristics associated with the use of public health services, such as older age and living in a household with few members, headed by an individual with no educational lag; a high socioeconomic status; and receiving cash transfers from social programs. Conversely, a high individual educational level was associated with reduced use of public health services.

### 3.2. Factors Associated with the Use of Public Outpatient Services

Table 3 shows that in both populations, being a woman increased the likelihood of using health services. This increase was 68% [95% CI 1.1, 2.4] for the IP compared to 63% [95% CI 1.4, 1.9] for the NIP. Meanwhile, having a job was associated with a lower probability of using health services among both the IP [OR = 0.48; CI 95% 0.3, 0.7] and the NIP [OR = 0.60; 95% CI 0.5, 0.7].

Having health insurance was the most important predictor of the use of public health services for both populations. However, this factor weighed more heavily in the IP: having SS coverage increased the likelihood of using public services by a factor of 14.57 [95% CI 6.3, 33.6] in this community, while for the NIP, the probability increased by 10.6 times [95% CI 8, 14.1]. Being affiliated with an insurance scheme aimed at the population without SS coverage increased the likelihood of using public health services by 5.6 times [95% CI 2.7, 11.6] for the IP and by 4.08 times [95% CI 3.1, 5.4] for the NIP, compared to those with no insurance.

We identified other factors that influenced the use of health services among the NIP. The age of individuals correlated positively with the use of health services, with the probability of using health services increasing as a person aged [OR = 1.02; 95% CI 1.01, 1.02]. Households that had received cash transfers from social programs were 25% [95% CI 1.0, 1.5] more likely to use public health services compared to those that had not received them. Lastly, households located in municipalities with very low marginalization rates were also more likely to use public health services [OR = 1.03; IC 95% 1.0, 1.1] compared to those located in highly marginalized municipalities.

### 3.3. Health Needs Motivating the Use of Public Outpatient Services

Table 4 shows that approximately one-third of the population aged ≥15 years who had used public outpatient services in the last month reported diabetes or hypertension as their main need, with similar figures for both the IP and NIP. However, although respiratory and intestinal infections were the second most reported health concerns for both groups, they were more prevalent in the IP than in the NIP (16.8% vs. 10%) (*p* < 0.05). The other non-communicable diseases, the third cause for seeking care in both groups, were more prevalent in the NIP (19.7%) than in the IP (13.8%) (*p* < 0.1).

The IP was 2.5 times more likely than the NIP to seek care as a result of experiencing non-specific symptoms, signs and abnormal findings (8.9% vs. 3.5%). In contrast, non-indigenous patients were three times more likely than indigenous patients to attend follow-up or control appointments.

## 4. Discussion

Our findings indicate that the indigenous population (IP) suffers from a significant socioeconomic lag compared to the non-indigenous population (NIP). We also found a lower rate of self-reported health needs and use of services among the IP, suggesting the presence of mechanisms that serve to exclude this community from utilizing these services. Previous studies have recognized geographical barriers such as greater distances to health facilities, as well as economic obstacles regarding direct and indirect costs, that reduce the likelihood of indigenous people using public health services [37]. However, the prevailing explanation based on studies of public health and health anthropology point to cultural factors as being responsible for the failed attempts to achieve harmony between the IP and public health services. These cultural factors pertain principally to communication problems resulting from linguistic differences and contrasting worldviews related to health/disease care processes. However, theoretical frameworks common in academic research have neglected to include the influence and effects of social stratification in health services, as well as its effects regarding discrimination against and mistreatment of indigenous people receiving healthcare [38,39]. Our results confirm the structural gap that hinders access of the IP to public health services. These structural factors contribute to the pronounced lag in highly marginalized rural regions, the low level of schooling, and the low rate of Social Security (SS) coverage. These sociodemographic characteristics are determined by a form of social organization that has systematically excluded the IP from various spheres of social life, and this largely explains the lack of healthcare available at public outpatient services.

The decision to use healthcare services begins with the recognition of a health need requiring care in a health facility, and our findings indicate that the IP is less likely to recognize such needs. This does not necessarily indicate that members of this community have fewer health problems, but rather that they do not recognize them as needs that must be addressed in public biomedical facilities [40]. However, even when indigenous people do recognize the need to use health services, they report obtaining less care from health personnel than their NIP counterparts, which may suggest discriminatory practices [5]. In addition, the IP makes greater use of services provided by traditional and alternative personnel, utilizing these services almost twice as often as the NIP. In fact, the frequency of use of outpatient services by the indigenous community with health needs is lower than that reported in another study which indicated a frequency of 87% for the total Mexican population with health problems in the previous month [41]. On the other hand, the decision to seek care from alternative personnel could reflect a predilection determined by cultural influences, since this personnel, as well as health assistants and midwives, often share the language and cultural perspective of the IP [15]. Research by Mexican health anthropologists has uncovered extensive evidence of the various care alternatives in addition to biomedical services that indigenous people turn to based on their cultural preferences and on the availability and ease of access to these services [37,39,40,42,43].

The results of our analysis are consistent with what has been reported in other studies. Having health insurance is the most important variable predicting the use of public health services [11,12,26,32,44]. It is worth noting the great importance of such coverage for the IP. This could be explained by the fact that the severe poverty and precarious living conditions in this community make them more dependent on insurance schemes to enable them to use public outpatient services [12,44]. However, even among the population with SS, one in three individuals in both the IP and NIP uses private services; 40% of the IP with coverage for those lacking SS makes use of private health services. This result, far from reflecting personal preferences, reveals the need for users to seek various options for attending to their health needs, even if this entails out-of-pocket expenses. The use of private services even by the poorest populations, including the IP, could be explained by the unavailability of public health services and the persistence of barriers to their use. From 2004 to 2010, the growth in the number of health facilities and physicians who care for the population without SS, including the majority of the IP, was not proportional to the growth 10.3 times that of affiliates utilizing these schemes during the same period. This contrasts with the sustained growth in the supply of doctors’ offices adjacent to private pharmacies (DAPPs) [41]. We must also consider the shortage of medicines, the long waiting times and the perceived low quality of public services -including those affiliated with SS- all factors that discourage their use [45].

Our results also point to the role of gender. In both the IP and NIP, women are the primary users of public health services. Other studies have documented that, compared to men, women are more aware of health problems [46], play a more active role as family caregivers and are more frequent users of health services [47,48]. In the case of Mexico, this situation could also be the result of social policies vigorously promoted from the late 1990s until December 2018. During these years, the PROSPERA government program (formerly *Progresa-Oportunidades*) figured prominently in encouraging women to use health services, mainly for reasons pertaining to sexual, reproductive and pediatric health [38,49].

The results of our study also document that being employed tends to reduce the use of public health services. This may be because work activities are mainly reported by men, who perceive fewer health problems and make less use of services [46,48]. In addition, those who are working often have less time to devote to health concerns. Moreover, members of the IP are often employed in agriculture or commerce, activities usually excluded from receiving SS benefits. This suggests the presence of forms of social and economic organization that discriminate against the IP.

It is noteworthy that in indigenous households, participating in government cash-transfer programs was not associated with greater use of public outpatient services. This could be explained by the fact that these households are generally located in rural and marginalized areas, with limited infrastructure and few health personnel [50]. It also helps explain why NIP households located in less marginalized municipalities make greater use of public outpatient services.

Our study found that diabetes and hypertension constituted the primary motive for using outpatient services. This contrasts with the results of the 2018-19 National Health and Nutrition Survey (ENSANUT), which found that these diseases were the second and third most common reasons for consultation among the Mexican population as a whole [51]. It is noteworthy that the need to attend to these conditions is one of the most common reasons for seeking care among the IP, which is indicative of a changing epidemiological profile [5]. On the other hand, respiratory and gastrointestinal infections, which in our study represented the second most common motive for consultation, constituted the first and third causes among the general population [51]. These differences could result from the fact that our study included an older population and excluded children (among whom gastrointestinal and respiratory infections are more common), while the 2018–2019 ENSANUT included the entire Mexican population [51]. The fact that the IP reports more non-specific symptoms and signs as reasons for seeking care, as opposed to specific diagnoses, may illustrate the epistemological disagreements and misunderstandings that characterize the relationship between the IP and biomedical services. It could also reflect the fact that health personnel do not provide sufficient information to indigenous patients. In addition, the low educational level prevailing among the IP, as well as linguistic and cultural differences, may render it more difficult for people in indigenous communities to understand health information. Consistent with other studies, our analysis shows that the IP attends fewer consultations for disease control and follow-up [5]. This is relevant, since diabetes and hypertension require close and continuous monitoring, not only to achieve more effective disease control, but also to detect comorbidities in a timely manner. This indicates the importance of providing more effective follow-up among the IP, especially for chronic–degenerative conditions.

Our study had limitations. First, the cross-sectional design of the survey did not provide evidence of causality. Second, information was lacking concerning the type of health problem or condition experienced by individuals who refrained from seeking care in spite of having reported health needs. This made it impossible to know what health needs did not lead to the use of services, and hence our study could not provide a comprehensive overview of the health needs of this population. The third limitation was the construction of a health needs indicator that combined the reporting of a health problem with a condition that led to the search for care. Although this limited comparability with other studies on service utilization that are based solely on the reporting of health problems [11,12,26], we considered it pertinent to include other conditions. Although these other conditions were not perceived as health problems, they favored the use of outpatient services and could be related to the use of preventive services. The dearth of information on the perception of the severity of health needs constituted another limitation, as the IP and NIP may have different perspectives, and the perception of severity is one of the main determinants for seeking and using health services. Finally, the small sample size of the members of the IP who both reported health needs and utilized services limited further analysis of health needs. Nonetheless, we believe that our findings contribute to documenting the fact that the prevalence of reported health needs in the IP is lower than in the NIP. They also help identify factors associated with the use of public outpatient services that are specific to this population. Our study highlights the need for quality health services adapted to the needs and characteristics of the IP, not only in the provision of healthcare, but also for more effective monitoring and control of chronic diseases which are already prevalent among this population.

## 5. Conclusions

It is crucial to craft strategies capable of fostering greater use of public health services by the populations with the greatest need, such as the IP. On the one hand, we believe that it is vital to ensure universal health coverage as a social right; our findings clearly demonstrate that health insurance is the most important predictor for the use of services for both the IP and NIP. Linking insurance coverage to employment has led to the segmentation of the health system, with a consequent deepening of inequality [52,53]. On the other hand, it is critical to ensure financial protection in health and reduce out-of-pocket expenses for the use of public and private services. This is just as important to achieving universal health coverage as increasing public spending on health [54]. Having universal health insurance will also contribute to achieving this goal.

In most utilization studies, variables pertaining to ethnicity are incorporated as independent or adjustment variables in the analyses and models, preventing a consideration of the specific characteristics of the IP. One contribution of this study is that it provides a specific analysis for each population, considering its most important individual, household and rural/urban area characteristics and exploring how these characteristics influence the use of health services by both the IP and NIP in a nationally representative sample.

## Figures and Tables

**Figure 1 ijerph-20-03048-f001:**
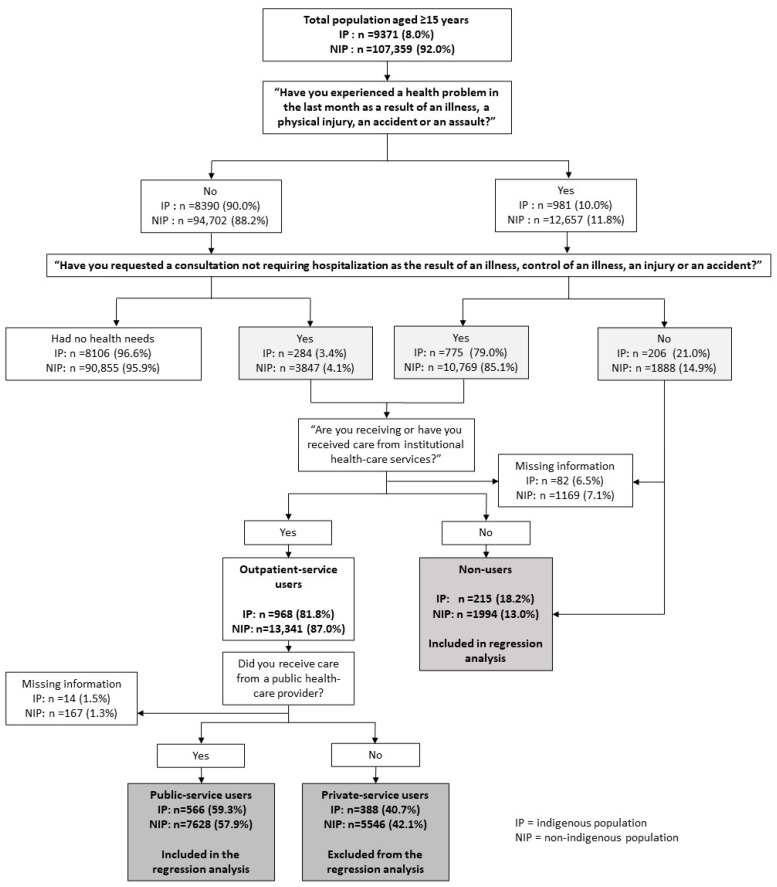
Analytical sample selection, ENSANUT 2018–2019.

**Figure 2 ijerph-20-03048-f002:**
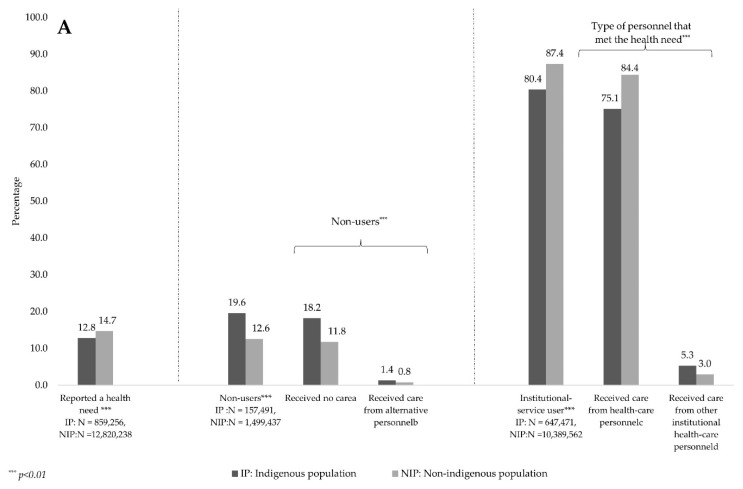
Prevalence of health needs, health personnel who provided care and institutions utilized, according to ethnicity, Mexico, ENSANUT 2018–2019. (**A**) presents the percentages of the IP and NIP aged ≥15 years that reported a health need in the last month. We identified those who refrained from using health services (that is, those who either received no care or received care from alternative personnel), as well as those who used institutional services (those who received care from a physician or other type of institutional personnel). (**B**) presents the types of institution utilized to obtain care, according to the insurance scheme with which users were affiliated. The IP and NIP were compared via chi-square tests, and proportions were estimated taking into account the survey design. Notes: a. Received no care from anyone or received care from a relative, friend or neighbor, and depended on a drugstore for treatment. b. Alternative personnel included herbalists, midwives, naturopaths and homeopaths. c. Health personnel included general practitioners and specialists. d. Other institutional health personnel included nurses, nutritionists, dentists and health assistants. e. Social Security schemes included the Mexican Social Security Institute (IMSS), the Institute of Social Security and Services for State Workers (ISSSTE), the health insurance carrier for the Mexican oil company, Petróleos de México (PEMEX) and Social Security Institutions for the Mexican Army and Navy. These schemes concentrate 23.4% of the IP and 54.6% of the NIP. f. Schemes for the population without SS coverage included the Seguro Popular and IMSS Prospera programs. These schemes concentrated 65.6% of the IP and 31.4% of the NIP. g. Private prepaid health insurance included 0.1% of the IP and 1.0% of the NIP.

**Table 1 ijerph-20-03048-t001:** Sociodemographic characteristics of the population aged 15 years and over, according to ethnicity, Mexico, ENSANUT 2018–2019.

	Total Population	Indigenous Population			Non-Indigenous Population			
n	116,730	9371			107,359			
Weighted N	94,146,910	6,724,738			87,422,172			
%	100.0%	7.1%			92.9%			
	Freq.	Freq.	%	[95% CI]	Freq.	%	[95% CI]	*p*-Value
Individual sociodemographic characteristics								
Sex								0.08
Male	55,577	4435	46.6	[45.9, 47.4]	51,142	47.3	[47.1, 47.6]	
Female	61,153	4936	53.4	[52.6, 54.1]	56,217	52.7	[52.4, 52.9]	
Age (years, continuous) ^©^	116,730	9371	40.5	[39.8, 41.2]	107,359	40.8	[40.6, 41.0]	0.37
Years of schooling								0.00
0	6571	1342	15.0	[13.8, 16.3]	5229	4.4	[4.20, 4.60]	
1–6 (elementary school or less)	28,133	3444	37.4	[35.7, 39.1]	24,689	22.3	[21.8, 22.7]	
7–9 (completed or incomplete middle school)	34,326	2522	26.1	[24.8, 27.5]	31,804	28.8	[28.4, 29.3]	
≥10 (high school and beyond	47,700	2063	21.5	[20.1, 23.1]	45,637	44.5	[43.8, 45.1]	
Civil status								0.00
Without partner	49,133	3512	37.4	[36.1, 38.7]	45,621	43.4	[42.9, 43.9]	
With partner	67,597	5859	62.6	[61.3, 63.9]	61,738	56.6	[56.1, 57.1]	
Health insurance schemes								0.00
None	19,507	1398	16.4	[14.8, 18.1]	18,109	19.2	[18.7, 19.8]	
Social Security ^a^	51,747	1689	15.7	[14.1, 17.4]	50,058	47.4	[46.7, 48.2]	
Schemes for those without Social Security coverage ^b^	43,812	6223	67.8	[65.4, 70.1]	37,589	32.1	[31.3, 32.8]	
Private or other ^c^	1116	17	0.1	[0.10, 0.30]	1099	1.3	[1.10, 1.40]	
Employed								0.00
No	49,987	4195	45.2	[43.8, 46.6]	45,792	42.6	[42.1, 43.0]	
Yes	66,743	5176	54.8	[53.4, 56.2]	61,567	57.4	[57.0, 57.9]	
Household characteristics								
Head of household with educational gap								0.00
No	65,130	2621	26.2	[24.3, 28.2]	62,509	59.2	[58.4, 60.0]	
Yes	51,600	6750	73.8	[71.8, 75.7]	44,850	40.8	[40.0, 41.6]	
Number of household members ^©^	116,730	9371	4.8	[4.70, 4.90]	107,359	4.2	[4.20, 4.20]	0.00
Socioeconomic status (tertiles)								0.00
Low	36,411	6342	70.7	[68.3, 73.0]	30,069	26.1	[25.3, 26.9]	
Intermediate	38,875	2219	21.2	[19.3, 23.2]	36,656	33.1	[32.3, 33.8]	
High	41,444	810	8.1	[6.70, 9.70]	40,634	40.8	[40.0, 41.7]	
Benefited from a social program								0.00
No	96,276	6070	65.1	[63.1, 67.1]	90,206	86.1	[85.7, 86.5]	
Yes	20,454	3301	34.9	[32.9, 36.9]	17,153	13.9	[13.5, 14.3]	
Area of residence								0.00
Rural (<2500 inhab.)	31,288	4748	49.4	[45.6, 53.2]	26,540	19.4	[18.9, 20.0]	
Urban	85,442	4623	50.6	[46.8, 54.4]	80,819	80.6	[80.0, 81.1]	
Municipal marginalization index ^©^	116,730	9371	53.1	[52.7, 53.6]	107,359	58.1	[58.0, 58.1]	0.00

This table compares the sociodemographic characteristics of individuals aged ≥15 years from the indigenous vs. the non-indigenous populations. Chi-square and Student’s *t* tests were used for categorical and continuous variables, respectively. Proportions and 95% CIs were estimated, taking the survey design into account. Notes **^©^** Continuous variables. ^a^ Social Security schemes included the Mexican Social Security Institute (IMSS), the Institute of Social Security and Services for State Workers (ISSSTE), the health insurance carrier for the Mexican oil company, Petróleos de México (PEMEX), and Social Security Institutions for the Mexican Army and Navy. ^b^ Health insurance schemes for those without Social Security coverage included the Seguro Popular and IMSS Prospera programs. ^c^ Private and others included private prepaid insurance carriers and others which were not specified.

**Table 2 ijerph-20-03048-t002:** Sociodemographic characteristics of the population aged 15 years and over who reported health needs and use of public outpatient health services, according to ethnicity, Mexico, ENSANUT 2018–2019.

	Indigenous Population						Non-Indigenous Population					
	Used Public Outpatient Health Services						Used Public Outpatient Health Services					
		Yes		No				Yes		No		
n		566		215				7628		1994		
Weighted N		355,059		157,491				5,675,898		1,499,437		
%		56.0% ^a,^***		19.6% ^b,^***				55.4% ^c,^***		12.6% ^d,^***		
Variables	n	%	[95% CI]	%	[95% CI]	*p*-Value	n	%	[95% CI]	%	[95% CI]	*p*-Value
Individual sociodemographic characteristics												
Sex						0.00						0.00
Male	336	38.6	[34.5, 42.9]	56.9	[51.2, 62.5]		3877	36.9	[35.5, 38.2]	53.7	[50.9, 56.4]	
Female	445	61.4	[57.1, 65.5]	43.1	[37.5, 48.8]		5745	63.1	[61.8, 64.5]	46.4	[43.6, 49.1]	
Age (years, continuous) ^©^	781	48.9	[46.9, 50.8]	46.7	[44.2, 49.1]	0.20	9622	50.4	[49.8, 51.0]	42.9	[42.0, 43.9]	0.00
Years of schooling						0.82						0.02
0	151	19.8	[16.3, 23.7]	18.4	[14.0, 23.8]		697	7.1	[6.4, 7.9]	5.9	[4.90, 7.20]	
1–6 (elementary school or less)	309	38.2	[33.5, 43.2]	42.2	[34.9, 49.8]		2818	30.3	[28.9, 31.8]	26.5	[24.2, 28.9]	
7–9 (completed or incomplete middle school)	182	23.2	[19.0, 27.9]	20.9	[15.4, 27.7]		2822	27.8	[26.5, 29.2]	30.0	[27.5, 32.7]	
≥10 (high school and beyond	139	18.9	[15.4, 22.9]	18.5	[13.7, 24.5]		3285	34.8	[33.2,36.3]	37.6	[34.9, 40.3]	
Civil status						0.34						0.14
Without partner	250	31.2	[27.1, 35.7]	35.1	[27.6, 43.5]		3695	38.8	[37.3, 40.4]	41.2	[38.5, 44.0]	
With partner	531	68.8	[64.3, 72.9]	64.9	[56.5, 72.4]		5927	61.2	[59.6, 62.7]	58.8	[56.0, 61.5]	
Health insurance schemes						0.00						0.00
None	51	2.7	[1.50, 4.90]	17.7	[13.1, 23.6]		590	3.3	[2.70, 4.10]	22.4	[20.0, 25.0]	
Social Security	206	28.1	[23.6, 33.0]	11.5	[7.6, 17.1]		5511	65.9	[64.3, 67.5]	34.5	[31.8, 37.2]	
Schemes for population without SS coverage	521	69.2	[64.2, 73.8]	70.7	[63.4, 77.1]		3483	30.7	[29.2, 32.2]	42.3	[39.4, 45.3]	
Private or other							17	0.1	[0.0, 0.2]	0.8	[0.30, 1.90]	
Employed						0.00						0.00
No	393	55.7	[50.6, 60.6]	35.1	[28.8, 41.9]		5006	56.7	[55.1, 58.2]	38.0	[35.4, 40.7]	
Yes	388	44.3	[39.4, 49.4]	64.9	[58.1, 71.2]		4616	43.4	[41.8, 44.9]	62.0	[59.3, 64.6]	
Household characteristics												
Head of household with educational lag						0.26						0.68
No	234	32.1	[27.1, 37.6]	27.2	[20.9, 34.5]		5295	55.9	[54.1, 57.6]	55.1	[52.2, 58.0]	
Yes	547	67.9	[62.4, 72.9]	72.8	[65.5, 79.1]		4327	44.1	[42.4, 45.9]	44.9	[42.0, 47.8]	
Number of members ^©^	781	4.2	[3.9, 4.5]	4.3	[3.90, 4.70]	0.69	9622	3.9	[3.8, 4.0]	4.0	[3.9, 4.2]	0.03
Socioeconomic status (tertiles)						0.17						0.00
Low	548	71.9	[67.4, 76.0]	78.1	[73.3, 82.1]		2998	25.8	[24.4, 27.3]	35.4	[32.6, 38.3]	
Intermediate	171	19.9	[16.3, 24.0]	17.4	[13.6, 22.0]		3569	38.1	[36.3, 39.9]	35.1	[31.9, 38.4]	
High	62	8.3	[6.4, 10.6]	4.5	[3.00, 6.70]		3055	36.1	[34.4, 37.8]	29.5	[26.7, 32.5]	
Benefited from a social program						0.89						0.00
No	441	57.1	[51.8, 62.2]	57.8	[50.2, 65.1]		7458	78.4	[77.1, 79.7]	84.6	[82.6, 86.4]	
Yes	340	42.9	[37.8, 48.2]	42.2	[34.9, 49.8]		2164	21.6	[20.3, 22.9]	15.4	[13.6, 17.4]	
Area of residence						0.03						0.00
Rural (<2500 inhab.)	390	45.8	[41.4, 50.2]	57.6	[52.9, 62.2]		2460	18.3	[17.2, 19.4]	23.6	[22.2, 25.0]	
Urban (≥2500 inhab.)	391	54.2	[49.8, 58.6]	42.4	[37.8, 47.1]		7162	81.7	[80.6, 82.8]	76.4	[75.0, 77.8]	
Municipal marginalization index ^©^	781	53.6	[53.1, 54.0]	52.2	[51.6, 52.9]	0.01	9622	58.1	[58.0, 58.2]	57.5	[57.4, 57.7]	0.00

This table presents the proportions of indigenous and non-indigenous individuals aged ≥15 years who reported health needs and received care from public outpatient healthcare providers compared to those who received no care whatsoever. It also shows the association between the sociodemographic characteristics of each population and the use of these services. Chi-square and Student’s *t* tests were used for categorical and continuous variables, respectively. Proportions and 95% CI were estimated, taking the survey design into account. The proportions of users and non-users of services in both populations were also compared using chi-square tests. Notes: *** *p* < 0.01. **^©^** Continuous variables. ^a^ For the IP, the proportion of public health service users was calculated in relation to outpatient service users with provider information (n = 954, N = 634,143). We lost 1.5% of observations due to lack of provider information. ^b^ Additionally, for the IP, the proportion of non-users of services was calculated in relation to the total number of individuals with health needs. (n = 1183, n = 804,962). We lost 6.5% of observations due to lack of information on whether or not care had been received. ^c^ For the NIP, the proportion of public health service users was calculated in relation to the total number of outpatient service users with provider information (n = 13,174, N = 10,236,742). We lost 1.3% of observations due to lack of provider information. ^d^ Additionally, for the NIP, the proportion of non-users was calculated in relation to the total number of individuals with health needs (n = 15,335, N = 11,888,999). We lost 7.1% of observations because of missing information on whether or not care had been received.

**Table 3 ijerph-20-03048-t003:** Factors associated with the use of public outpatient services among individuals ≥15 years old with health needs, according to ethnicity, Mexico, ENSANUT 2018-19.

Population	Indigenous ^a,Δ^			Non-Indigenous ^b,Δ^		
Non-Users of Outpatient Services (Reference)	n = 214			n = 1984		
Users of Public Outpatient Services	n = 564			n = 7617		
	OR	[95% CI]	*p*-Value	OR	[95% CI]	*p*-Value
Sociodemographic characteristics of users						
Sex						
Male	Ref.			Ref.		
Female	1.68	[1.14, 2.47]	0.01	1.63	[1.40, 1.90]	0.00
Age (years, continuous) ^©^	1.00	[0.98, 1.02]	0.99	1.02	[1.01, 1.02]	0.00
Age squared (years)	1.00	[1.00, 1.00]	0.86	1.00	[1.00, 1.00]	0.50
Categorical educational levels Years of schooling						
0	Ref.			Ref.		
1–6 (elementary school or less)	0.87	[0.50, 1.51]	0.62	0.98	[0.72, 1.35]	0.91
7–9 (completed or incomplete middle school)	1.00	[0.46, 2.20]	0.99	1.03	[0.72, 1.46]	0.89
≥10 (high school and beyond)	0.75	[0.32, 1.74]	0.50	0.95	[0.66, 1.38]	0.79
Civil status						
Without partner	Ref.			Ref.		
With partner	1.31	[0.80, 2.14]	0.28	1.04	[0.88, 1.22]	0.64
Employed						
No	Ref.			Ref.		
Yes	0.48	[0.31, 0.75]	0.00	0.60	[0.51, 0.70]	0.00
Health insurance schemes						
None	Ref.			Ref.		
Social Security	14.57	[6.32, 33.56]	0.00	10.63	[8.02, 14.09]	0.00
Schemes for the population without Social Security coverage	5.61	[2.71, 11.61]	0.00	4.08	[3.06, 5.44]	0.00
Private and other	n/a			0.86	[0.25, 3.01]	0.82
Household characteristics						
Head of household with educational lag						
Yes	Ref.			Ref.		
No	1.00	[0.58, 1.72]	1.00	1.05	[0.87, 1.28]	0.59
Number of household members	1.01	[0.91, 1.12]	0.85	1.03	[0.99, 1.07]	0.21
Socioeconomic status (by tertile)						
Low	Ref.			Ref.		
Intermediate	0.76	[0.42, 1.35]	0.35	1.11	[0.90, 1.37]	0.31
High	1.04	[0.53, 2.03]	0.91	0.99	[0.77, 1.29]	0.96
Affiliated with a social program						
No	Ref.			Ref.		
Yes	0.89	[0.54, 1.49]	0.66	1.25	[1.03, 1.53]	0.03
Area of residence						
Rural (<2500 inhab.)	Ref.			Ref.		
Urban (≥2500 inhab.)	1.29	[0.77, 2.16]	0.33	0.99	[0.83, 1.19]	0.95
Municipal marginalization index (continuous)	1.03	[0.98, 1.08]	0.23	1.03	[1.00, 1.07]	0.07

Logistic models were adjusted for indigenous and non-indigenous populations. The factors influencing the likelihood of using public outpatient services in each type of population were analyzed, and the base category in both models was non-users. Estimates took the survey design into account. Logistic model goodness-of-fit tests: in the IP, F-adjusted = 1.106 and *p* = 0.360; in the NIP, F-adjusted = 1.571 and *p* = 0.118. Notes: **^©^** Continuous variables. ^Δ^ We lost three observations in the IP (0.4%) and 21 in the NIP (0.2%) because of missing information on insurance coverage. ^a^ Percentage of the IP served by type of public healthcare provider. Social Security schemes included the Mexican Social Security Institute-IMSS: 20.26%; the Institute of Social Security and Services for State Workers-ISSSTE: 4.94%; the health insurance carrier for the Mexican oil company, Petróleos de México-PEMEX: 0.28%; and Social Security Institutions for the Army and the Navy: 1.49%. Institutions that served the population without Social Security coverage included State Health Services: 65.98% and the IMSS Prospera program: 7.06%. ^b^ Percentage of NIP served by type of public healthcare provider. Social Security schemes included the Mexican Institute of Social Security-IMSS: 51.04%; the Institute of Social Security and Services for State Workers-ISSSTE: 11.63%; the health insurance carrier for the Mexican oil company, Petróleos de Méxic-PEMEX: 0.56%; and Social Security Institutions for the Army and the Navy: 0.90%. Institutions that served the population without Social Security coverage included State Health Services: 34.88% and the IMSS Prospera program: 0.99%.

**Table 4 ijerph-20-03048-t004:** Health needs reported by users of public outpatient services, by ethnicity, Mexico, ENSANUT 2018–2019.

Population	Indigenous				Non-Indigenous				
N	249				3570				
Weighted N	146,881				2,571,065				
%	41.4% ^a^				45.3% ^b^				
	N	n	%	[95% CI]	N	n	%	[95% CI]	*p*-Value
Respiratory and gastrointestinal infections	24,707	37	16.8	[12.7, 22.0]	257,214	405	10.0	[8.9, 11.2]	0.012
Other infectious diseases	6090	10	4.2	[2.5, 6.8]	68,307	113	2.7	[2.1, 3.4]	0.246
Diabetes and hypertension	40,414	75	27.5	[22.8, 32.7]	776,400	1037	30.2	[28.1, 32.3]	0.442
Cardiovascular and circulatory diseases	3380	6	2.3	[1.0, 5.2]	88,470	121	3.4	[2.7, 4.3]	0.412
Non-infectious gastrointestinal disease and symptoms	14,229	23	9.7	[6.0, 15.4]	181,425	250	7.1	[6.0, 8.3]	0.236
Other non-communicable diseases	20,243	31	13.8	[10.3, 18.2]	505,719	684	19.7	[17.9, 21.6]	0.080
Musculoskeletal diseases	7229	13	4.9	[2.6, 9.0]	158,932	226	6.2	[5.2, 7.3]	0.471
Mental and behavioral disorders	1524	4	1.0	[0.5, 2.1]	36,392	53	1.4	[1.0, 2.0]	0.562
Pregnancy, childbirth and puerperium	6437	11	4.4	[2.3, 8.3]	116,068	171	4.5	[3.7, 5.4]	0.930
Accidental injuries and assaults	7804	11	5.3	[3.7, 7.5]	188,302	252	7.3	[6.2, 8.6]	0.503
Symptoms, signs and abnormal findings	13,030	23	8.9	[5.7, 13.6]	90,245	125	3.5	[2.8, 4.4]	0.000
Consultation for control or follow-up of results	1794	5	1.2	[0.5, 3.2]	103,591	133	4.0	[3.1, 5.1]	0.010

This table compares the types of health needs reported by the outpatient service users from the IP and NIP. Differences were assessed through chi-square testing. Estimates took the survey design into account. Notes: ^a^ Proportion of the indigenous population that used public outpatient services (n = 566) (N = 355,059). ^b^ Proportion of the non-indigenous population that used public outpatient services (n = 7628) (N = 5,675,898).

## Data Availability

Data can be made available upon request.

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
