# Peer review of "Use of Outpatient Health Services by Mexicans Aged 15 Years and Older, According to Ethnicity"

_ijerph, 2023, doi:10.3390/ijerph20043048_

Round 1

Reviewer 1 Report

Thank you very much for allowing me review your study about the “Use of Outpatient Health Services by Mexicans Aged 15 Years and Older, According to Ethnicity”. It is an interesting and well conducted analysis that is needed in order to detect population needs (including minorities) to developed tailored interventions. Please found some minor comments below:

·         The paragraph « Data and population » could be named “Study design, data and population”.

·         Why are people requiring a doctor’s appointment that ended in hospitalization excluded? What is the estimate percentage of that population?

·         Were pharmacists considered in the classification of care providers?

·         Figures resolution needs to be improved as it is hard to read.

·         It would be recommended that Figure 2 includes n (as well as %) to facilitate comprehension.

·         When comparing government programs specifically designed for individuals without SS coverage between both groups it would be recommended to avoid mixing categories, therefore it would be “(67.8% vs. 32.1%)” rather than “(67.8% vs. 47.4%)”.

·         Table 2 needs to state each group (IP and NIP).

·         Correct “PI” in line 334.

·         Table 4 includes health needs reported by users of public outpatient services; nevertheless, that variable was not included or described in the methodology section. It should be included by detailing how it was recorded (patients’ report or diagnose from health care professionals) (list of health problems or open questions)?

·         How was the sample of outpatient-service users identified and selected? It would be recommended to include it in the methods section.

Author Response

Thank the reviewer for giving us the opportunity to improve our paper. The new version of the article has been submitted marking changes with yellow to facilitate the identification of our responses to the comments.

In the attached file we provide a point-by-point response to each of the observations.

Reviewer 2 Report

The article titled “Use of Outpatient Health Services by Mexicans Aged 15 Years and Older, According to Ethnicity” by Pelcastre-Villafuerte et al is a systematic analysis of the relationship between socio-demographic factors and the utilization of healthcare services by indigenous and non-indigenous populations in Mexico. This is an interesting study given the need for equity in healthcare for the betterment of population health. A major issue here is that many literature citations were in Spanish and, therefore, not assessable. This limits the ability to evaluate novelty of the study and identify any correlations with other previously conducted studies. Secondly, both figures are blurry and not clear enough to read and evaluate.

Other comments related to data analyses:

1.     In the statement on lines 71-75 about insurance coverage for IP, the percentages add up to more that 100%.

2.     The numbers for n (from survey) and N (weighted based on population) do not seem to be accurate e.g. 9,371 is 8% of 116,730 instead of the 7.1% indicated based on the N values.

3.     In Table 2, Column headers indicating IP and NIP are missing. Also, it is not clear why the total percent for the Yes and No columns are not adding up to 100%. What population is being excluded from this comparison? If a certain population is being excluded, then is this the right comparison? 

Throughout the paper, all numbers and percentages for assessed population, weighted population and specific service utilizers should be verified.

Author Response

(The authors gave the same response as above.)
